# SONNET: Solar-disaggregation-based Day-ahead Probabilistic Net Load Forecasting with Transformers

## Abstract

The global transition towards sustainable energy sources has positioned solar power as a cornerstone of modern electricity systems, underscoring the critical need for advanced forecasting techniques in grid management. Accurate net load forecasting is crucial for efficient and reliable power grid operations, especially with the rapid deployment of behind-the-meter (BTM) renewable energy sources such as rooftop solar. Notably, BTM solar generation is neither controlled nor monitored by utilities and hence only net load data are observed. Different from load forecasting, net load forecasting faces new challenges because BTM solar, a major component of net load, behaves very differently from and is much more variable than loads. To exploit the distinct natures of solar generation and load and unlock their predictive potentials, we propose **SONNET**, which stands for **SO**lar-disaggregatio**N**-based **NE**t load forecasting with **T**ransformers. It is a novel probabilistic net load forecasting method based on disaggregating net loads into solar generation and loads and feeding both into the predictors. The method further features a) an enhanced Transformer architecture that integrates both historical and future input data, employing a combination of self-attention and cross-attention mechanisms, and b) a data augmentation method that enhances the robustness of net load forecasts against weather forecast errors. Extensive experiments are conducted based on the comprehensive real-world data set from a recent net load forecasting competition organized by the U.S. Department of Energy (DOE). It is demonstrated that our proposed method both improves the accuracy and reduces the uncertainty of net load forecasts. Notably, our proposed method significantly outperforms the state-of-the-art. The proposed techniques also have broad applications for energy and/or general forecasting-related problems.

## 1 Introduction

In an electric grid, the net load is the difference between the electric load and the "behind-the-meter" (BTM) power generation, notably BTM solar generation such as rooftop solar on residential, commercial, and industrial premises. As grid operators typically neither monitor nor control BTM generation, net load (as opposed to load) is what grid operators need to procure energy supply for at all times. With the rapid growth of renewable energy sources in our power systems, net load forecasting plays an increasingly crucial role for grid operators to efficiently and reliably plan their daily energy procurement. The importance of net load forecasting is no less exemplified by the first net load forecasting competition recently organized by the U.S. Department of Energy (DOE) (HeroX, 2023), anticipating a very significant penetration of BTM solar generation in the coming years.

Traditionally, with no or little BTM solar generation, the problem of net load forecasting reduces to load forecasting only, for which there have been decades of ongoing innovations and practices (Kuster et al., 2017). As BTM solar penetration increases, the corresponding shift in the composition of net load leads to a fundamental change in the problem nature of net load forecasting: While a) electric load in a future time slot correlates with both historical loads and future weather (among other factors potentially), b) solar generation is predominantly determined by meteorological conditions, particularly solar irradiance (Ahmed et al., 2020). Given these, an important practical limitation that makes net load forecasting a particularly challenging and new problem is the following: It is

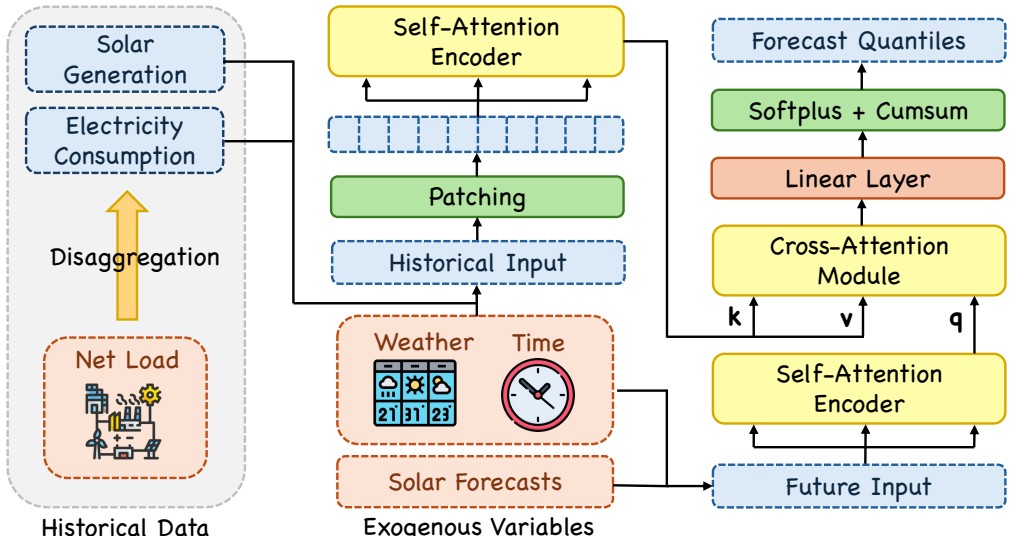

Figure 1: Overall structure of SONNET.

almost always the case that *only the net load data traces are observed and available to grid operators and utilities, not the separate traces of load and solar generation.* Thus, while methods for *load* forecasting may directly be implemented on *net load* traces (Hong and Fan, 2016), their effectiveness could degrade as *solar* generation, a major component of net load, is of a very different nature and much more variable than load. A critical question is thus the following: Given only net load data, how can the very different natures of load and solar generation both be exploited to achieve accurate and robust net load forecasting?

**Contributions:** We propose SONNET: **SO**lar-disaggregatio**N**-based **NE**t load forecasting with **T**ransformers (cf. Fig. 1). SONNET produces day-ahead probabilistic net load forecasting that is not only accurate but also robust to weather forecast errors. The main contributions are as follows:

• We develop a fully unsupervised BTM solar disaggregation algorithm that estimates solar generation traces given only net load traces, achieving performance close to that of supervised learning given ground truth solar generation. Both the disaggregated solar generation and loads are then fed as part of the input data to the predictors for net load forecasting.

• We develop a Transformer-based architecture that integrates both historical and future input data, employing a combination of self-attention and cross-attention mechanisms to enhance the accuracy of probabilistic net load forecasting by leveraging exogenous variables such as weather forecasts.

• We develop a physical-model-based data augmentation method that a) improves the predictors' robustness to weather forecast errors, while also b) alleviating the issue of a limited amount of training data, a common issue in practice.

• We conduct extensive experiments on *day-ahead* probabilistic net load forecasting, a particularly challenging and important problem in power systems. We demonstrate a significant performance gain by our proposed method over the state-of-the-art based on the comprehensive real-world data.

## 2 RELATED WORK

**Net Load Forecasting:** Net load forecasting has gained increasing attention in recent years due to the growth of BTM solar generation. Algorithms have been developed that take net loads and weather features, including solar irradiance, as input, and forecast net loads as output (see, e.g., (Tziolis et al., 2023; Faustine and Pereira, 2022; Zhang et al., 2023) among others.) These works, however, do not explicitly exploit the different natures of the underlying solar generation and load processes. Other works took the approach of estimating load and solar generation separately. Most of these works require knowledge of the ground truth solar generation and load traces as supervised labels for training their respective predictors (see (Zhang et al., 2021; Alipour et al., 2020) among others.)

As separately monitored solar generation and load traces are typically not available in practice, disaggregation-based methods have been proposed that only rely on net load traces which are indeed available. (Wang et al., 2018) decomposes net loads into loads, solar generation, and residuals, albeit with a relatively simple disaggregation algorithm and evaluated on synthetic data only. (Jia et al., 2023) decomposes net load into load and solar generation. These works train separate predictors for the respective sub-traces albeit with relatively simple predictor architectures. Notably, while different techniques have been proposed, there has not been a verifiable state-of-the-art due to the lack of a common testing benchmark, until the net load forecasting competition recently organized by the U.S. DOE (HeroX, 2023). In this competition, more than 90 teams have participated including both researchers and many commercial forecasting companies. The top-ranked results based on a common set of real-world data and evaluation metrics demonstrate the first verifiable state-of-the-art for net load forecasting to the best of the authors' knowledge.

**Time Series Forecasting with Transformers:**  Transformers, introduced in (Vaswani et al., 2017), are highly effective in sequence modeling benefiting from the attention mechanism. Researchers have successfully applied Transformers to time series forecasting across various fields (Wen et al., 2022; Liang et al., 2024; Wang et al., 2023b), including energy consumption (Nazir et al., 2023), traffic (Zhang et al., 2024), and finance (Mulvey et al., 2022). Models like (Zhou et al., 2021; Wu et al., 2021; Zhou et al., 2022) modify the underlying attention mechanism to better accommodate long-term series forecasting and reduce computational complexity. Alternatively, another approach avoids altering the attention mechanism itself and instead employs the original Transformer block for time series modeling (Nie et al., 2022; Zhang and Yan, 2022; Liu et al., 2023).

## 3   PROBLEM FORMULATION

For an electricity-consuming region of interest, its net load at time $t$ is denoted as $N_t$. Based on the common practice of power system operations, it is crucial to perform net load forecasting on a *day-ahead* schedule. Specifically, we adopt the schedule from the DOE competition which broadly embodies real-world power system operation requirements: At 10 am every day, the hourly net loads of the 24 hours of the *next* day need to be forecasted. Moreover, due to the risk management requirement of reliable power system operations, *probabilistic* forecasts are needed. In other words, the goal is to provide a probability distribution of each hourly net load of the next day. To evaluate an estimated probability distribution based on the observed ground truths, a commonly used metric (also adopted in the DOE competition) is the Continuous Ranked Probability Score (CRPS):

$$\text{CRPS} = \frac{1}{n} \sum_{i=1}^{n} \int \left( F_i(x) - O_i(x) \right)^2 \, dx, \quad O_i = \begin{cases} 0, & \text{if } x < x_i \\ 1, & \text{if } x \geq x_i \end{cases} \tag{1}$$

where $n$ is the number of forecasted hours, $F_i(x)$ is the Cumulative Distribution Function (CDF) of the forecast quantity $x$ at hour $i$, $O_i(x)$ is the "ideal" staircase CDF given the observed value $x_i$. Importantly, CRPS captures both a) the mean's accuracy, and b) the uncertainty/confidence of the estimated probability distribution (Matheson and Winkler, 1976; Gneiting and Raftery, 2007).

## 4   SONNET: THE METHODOLOGY

### 4.1   BTM SOLAR DISAGGREGATION

The net load $N_t$, load $L_t$, and BTM solar generation $G_t$ satisfy the following simple relation:

$$N_t = L_t - G_t, \quad t = 1, ..., T. \tag{2}$$

While only the net load data are observed in practice, intuitively, having both the load and solar generation data separately can offer more information to facilitate net load forecasting. We aim to disaggregate each net load data trace $N_t$ into two data traces of solar generation $G_t$ and load $L_t$, and then utilize patterns from both traces to forecast future net loads.

In addition, while utilities do not monitor BTM solar generation $G_t$, they typically have some estimate of the installed solar capacity $C$ at a regional level. This is because solar energy interconnection in a power distribution system must be approved by utilities for reliability reasons, making solar capacity

information available to them. As such, the solar disaggregation problem is formulated as follows: For a region of interest, given a net load trace $N_t = L_t - G_t, \forall t$, and a rough estimate of the solar capacity $C$, estimate the underlying solar generation trace $G_t$ and load trace $L_t, \forall t$. For this, we develop a BTM solar disaggregation algorithm inspired by the general framework of (Pu and Zhao, 2023).

### 4.1.1 PHYSICAL MODEL

We introduce a general physical model of a solar system's energy production. A solar panel's generation can be estimated using a physical model that depends on several factors, including physical model parameters and related weather variables. (Wang et al., 2018; Pu and Zhao, 2023).

$$G_t \approx C \frac{I_{PV,t}}{I_{ref}} \left[ 1 - \mu(T_{PV,t} - T_{ref}) \right], \tag{3}$$

where $I_{PV,t}$ is the solar irradiance received on the solar panels, $C$ is the capacity of the solar panel, $\mu$ is the temperature coefficient, $T_{PV,t}$ is the cell temperature, and $T_{ref}$ and $I_{ref}$ are reference temperature and irradiance, respectively. The irradiance on panel $I_{PV,t}$ depends on direct normal irradiance (DNI) $I_{0,t}$, diffuse horizontal irradiance (DHI) $I_{d,t}$, and direct horizontal irradiance $I_{b,t}$:

$$I_{PV,t} = I_{0,t}\tau_{b,t}(\sin\alpha\cos\beta + \cos\alpha\sin\beta\cos(\gamma - A))$$
$$+ I_{d,t}\left(\frac{1 + \cos\beta}{2}\right) + (I_{b,t} + I_{d,t})\rho_t\left(\frac{1 - \cos\beta}{2}\right). \tag{4}$$

$\beta$ and $\gamma$ are the tilt angle and azimuth angle of the solar panels which are typically unknown to the utilities and need to be estimated. The meaning of other physical quantities and more details are relegated to Appendix A.2.1. In short, we denote the overall physical model of solar generation by $G_t = f(\boldsymbol{x}(t); \boldsymbol{\theta})$, where $\boldsymbol{\theta}$ contains the two *unknown* panel-dependent model parameters, tilt angle $\beta$ and azimuth angle $\gamma$, and $\boldsymbol{x}(t)$ contains all the external relevant variables that are measured (essentially weather, time, and location). While this physical model is formulated for an individual solar panel, we will apply it as an approximate model for a collective set of solar panels in a region.

### 4.1.2 SOLAR DISAGGREGATION

In principle, the parameter vector $\boldsymbol{\theta}$ containing the tilt angle and azimuth angle can be solved from just two equations given the solar power generation $G_{t_i}$ at two time instants $t_1, t_2$:

$$f(\boldsymbol{x}(t_i); \boldsymbol{\theta}) = G_{t_i}, \quad i = 1, 2 \tag{5}$$

However, when the solar generation $G_t$ is unknown and needs to be estimated, we must rely on other available data such as the net load data $N_t$ and additional input $\boldsymbol{x}(t)$. In particular, if the loads at two different time instances, $t$ and $t'$, are equal, we have

$$L_t = N_t + G_t = N_{t'} + G_{t'} = L_{t'} \tag{6}$$
$$\Rightarrow N_t + f(\boldsymbol{x}(t); \boldsymbol{\theta}) = N_{t'} + f(\boldsymbol{x}(t'); \boldsymbol{\theta}) \tag{7}$$

If the external variables $\boldsymbol{x}(t)$ and $\boldsymbol{x}(t')$ are not identical, the above equation forms a non-trivial equation of $\boldsymbol{\theta}$, as all other quantities are measured. As such, we seek to find a sufficiently large number of such equations that approximately hold to determine $\boldsymbol{\theta}$ without relying on any knowledge of the BTM solar generation $G_t$. Given only net load data $N_t$, while we cannot be certain that $L_t$ at different times are exactly the same, predictions can be made when they are likely to be sufficiently similar. Collecting a large dataset of similar load instances, named $\mathcal{T}$, allows us to learn $\boldsymbol{\theta}$ by forming an over-determined set of equations. Accordingly, a loss term to capture the load similarity within selected time pairs is defined as:

$$L^{(1)} = \sum_{(t_i, t_i') \in \mathcal{T}} \left( (N_{t_i} + f(\boldsymbol{x}(t_i); \boldsymbol{\theta})) - (N_{t_{i'}} + f(\boldsymbol{x}(t_i'); \boldsymbol{\theta})) \right)^2. \tag{8}$$

Additionally, two regularization terms are defined to penalize estimation inaccuracies, in particular, when either the estimated solar generation or the estimated loads (i.e., consumption) become negative:

$$L^{(2)} = \sum_{t=1}^{T} \max\left(-f(\boldsymbol{x}(t_i); \boldsymbol{\theta}), 0)\right)^2, \quad L^{(3)} = \sum_{t=1}^{T} \max\left(-(f(\boldsymbol{x}(t_i); \boldsymbol{\theta}) + N_{t_i}), 0)\right)^2. \tag{9}$$

Finally, we solve the following problem and estimate the physical model parameters $\boldsymbol{\theta}$ for the region:

$$\min_{\boldsymbol{\theta}} L^{(1)} + \gamma L^{(2)} + \eta L^{(3)}, \tag{10}$$

where $\gamma$ and $\eta$ are weights that balance the three losses.

---

**Algorithm 1** Similarity-based BTM Solar Disaggregation

---

1: **Input:** $N_t$, $\boldsymbol{x}(t)$, monthly solar capacity $C$
2: Initialize a set of time slot pairs $\mathcal{T}$. See Appendix A.2.2 for details.
3: Initialize $Loss_{min} = \infty$, $\boldsymbol{\theta}_{out}$, model parameters $\boldsymbol{\theta}$
4: **for** step in 1 to maxiter **do**
5:     **for** epoch in 1 to maxepochs **do**
6:         Given $\mathcal{T}$, calculate $Loss = L^{(1)} + \gamma L^{(2)} + \eta L^{(3)}$
7:         Update model parameters $\hat{\boldsymbol{\theta}}$ by backpropagation
8:     **end for**
9:     **if** $Loss_{min} > Loss$ **then**
10:         $Loss_{min} = Loss$, $\boldsymbol{\theta}_{out} = \hat{\boldsymbol{\theta}}$
11:     **else**
12:         Break
13:     **end if**
14:     Update solar estimation $\hat{G}_t = f(\boldsymbol{x}(t); \hat{\boldsymbol{\theta}})$ and load estimation $\hat{L}_t = f(\boldsymbol{x}(t); \hat{\boldsymbol{\theta}}) + N_t$
15:     Re-select $M$ neighboring time slot pairs with the most similar estimated loads $\hat{L}_t$.
16: **end for**
17: **Output:** Estimated physical model parameters $\boldsymbol{\theta}_{out}$

---

**Training Set Selection**: To collect time pairs with similar loads, we propose an iterative selection algorithm. Initially, we set the time pairs to be some neighboring pairs during daytime with a sufficiently large size, denoted by $M$, leveraging the fact that loads in close time slots tend to be similar due to the regularity of aggregate human behaviors. In subsequent iterations, neighboring time pairs in the training set are refined based on the most similar load pairs from the latest round of disaggregation. The detailed steps are provided in Algorithm 1.

## 4.2 PREDICTOR ARCHITECTURE

Given the disaggregated solar generation and load traces, we propose a Transformer-based architecture designed for net load prediction, as depicted in Figure 1. This architecture leverages both historical and future input data to enhance the accuracy of the forecasts, employing a combination of self-attention and cross-attention mechanisms.

**Exogenous Variables**: We integrate not only the net load variables $X \in \mathbb{R}^{n \times l}$ (net load, load, disaggregated solar generation), but also introduce exogenous variables $S \in \mathbb{R}^{m \times l}$ to assist in the net load forecasting. These exogenous variables encompass weather features $W$ (temperature, humidity, DHI, DNI), and time features $Z$. Here $l$ denotes the context window, $n$ denotes the number of net load variables, and $m$ denotes the number of exogenous variables. Detailed descriptions of these features are provided in the Appendix A.7.1.

**Historical and Future Input:** Our model processes both historical and future variables for net load forecasting. Specifically, the historical input $H$ includes both historical $X$ and $S$, such that $H = \texttt{Concat}(X, S) \in \mathbb{R}^{(m+n) \times l}$, while the future input consists of a) $\hat{S} \in \mathbb{R}^{m \times \hat{l}}$, i.e., forecasted exogenous variables, and b) $\hat{G} \in \mathbb{R}^{1 \times \hat{l}}$, i.e., forecasted solar generation computed based on the physical model (cf. Section 4.1.1) and the weather forecasts, where $\hat{l}$ denotes the prediction length.

For the historical input time series, we apply patching, which involves segmenting the time series into subseries-level patches that serve as input tokens to the Transformer. This technique has been demonstrated to enhance the effectiveness of capturing dynamic local patterns and semantic information while reducing computational costs in time series analysis (Nie et al., 2022). Given the patch length $p$ and stride $s$, the patch number of input series is given by $r = \lfloor (l - p)/s \rfloor + 1$, which indicates the sequence length after patching operation. Furthermore, we reshape the tensor

to merge the dimension of patch and feature, which will be $\tilde{H} = \texttt{Patching}(H) \in \mathbb{R}^{g \times r}$, where $g = p(m + n)$ is the generalized feature dimension. Conversely, the future input time series $\hat{S}$ retains its original resolution to ensure the prediction granularity aligns with our forecasting target.

**Self-Attention Encoder:** To process both historical and future inputs, we employ a self-attention encoder. This encoder is adept at modeling intra-sequence patterns and generates useful embeddings for subsequent stages. The self-attention encoder on input data can be represented as:

$$\texttt{SelfAttention}(\boldsymbol{x}) = \texttt{MHA}(\boldsymbol{x}', \boldsymbol{x}', \boldsymbol{x}'), \qquad (11)$$

where $\boldsymbol{x}' = \texttt{PE}(\boldsymbol{x})$, $\texttt{PE}$ stands for position embedding which we explain in details in Appendix A.3, and $\texttt{MHA}(x^Q, x^K, x^V)$ denotes the multi-head attention module in standard Transformer architecture with input $x^Q, x^K, x^V$ for query, key and value respectively (Vaswani et al., 2017), which we explain in details in Appendix A.4. Thus, we could obtain the historical embedding $\mathbf{E} = \texttt{SelfAttention}(\tilde{H})$ and future embedding $\hat{\mathbf{E}} = \texttt{SelfAttention}(\hat{S})$. By mixing all the features and learning the temporal patterns of the embeddings, the model can generate useful representations with the information from all variables.

**Cross-Attention Module:** Subsequent to obtaining $\mathbf{E}$ and $\hat{\mathbf{E}}$ through self-attention, we utilize a cross-attention module to model the relationship between future and historical data. The historical embedding serves as inputs of keys and values, while the future embedding acts as input of queries. The cross-attention mechanism calculates the relationship between each future day's embedding and the historical embeddings:

$$\texttt{CrossAttention}(\mathbf{E}, \hat{\mathbf{E}}) = \texttt{MHA}(\hat{\mathbf{E}}, \mathbf{E}, \mathbf{E}). \qquad (12)$$

This enables the historical sequence to be combined as a reference for future forecasting, enhancing the model's predictive capabilities by integrating past trends with expected conditions. Such a mechanism ensures that when weather-related forecasts are fed into the model, the model can look up similar conditions from the past for better prediction.

**Probabilistic Forecasting:** The embedding features derived from the cross-attention module $\mathbf{E_f} = \texttt{CrossAttention}(\mathbf{E}, \hat{\mathbf{E}}) \in \mathbb{R}^{d \times \hat{l}}$ are processed through a final linear layer, which is responsible for generating multiple quantiles at each time step for probabilistic forecasting. The linear layer $\mathbf{W}_f \in \mathbb{R}^{q \times d}$, where $q$ denotes the number of quantiles, transforms $\mathbf{E_f}$ into quantile predictions $\mathbf{Q} = \mathbf{W}_f \mathbf{E_f} \in \mathbb{R}^{q \times \hat{l}}$. To ensure non-negativity and monotonicity among the quantiles, we apply a softplus activation function $\sigma(\cdot)$ to all quantiles except the first one. This can be expressed as:

$$\mathbf{Q}'_i = \begin{cases} \mathbf{Q}_i & \text{if } i = 0, \\ \sigma(\mathbf{Q}_i) & \text{if } i > 0, \end{cases} \qquad (13)$$

where $\mathbf{Q}'_i$ denotes the adjusted $i$-th quantile for $i = 0, ..., q$. Then a cumulative sum operation is performed across the quantiles to ensure their ordered sequence: $\mathbf{Q}''_i = \sum_{j=1}^{i} \mathbf{Q}'_j$, where $\mathbf{Q}''_i$ represents the forecast of $i$-th quantile, maintaining a non-decreasing order across all time steps. Finally, we apply CRPS loss (the implementation is in Appendix A.5) between $\mathbf{Q}''$ and the ground truth $\mathbf{O}$ as described in equation 1 to train the model.

## 4.3 DATA AUGMENTATION

Although the ground truth data for historical net load and weather are available, importantly, *day-ahead* forecasts of certain weather variables can suffer from poor accuracy. This issue is particularly pronounced in areas such as Hawaii due to its fast-varying clouds. Thus, the trained predictors' robustness to weather forecast errors is a major concern, especially with a high level of penetration of solar generation. To improve the predictor's robustness to weather forecast errors, we design the following data augmentation steps for the training and validation data sets:

- **Error Simulation in Meteorological Features**: We introduce variability into each meteorological feature to simulate forecast errors. For irradiance features (e.g., Direct Normal Irradiance (DNI) and Diffuse Horizontal Irradiance (DHI)), we adjust the original value with a multiplicative error term $(1 + \epsilon)$, where $\epsilon$ is sampled from a normal distribution $N(0, \sigma_1)$ with a sufficiently large $\sigma_1$. For other weather features (e.g., temperature and humidity), we generate and apply additive

errors from the empirical forecasting error distributions $N(0, \sigma_2)$ (Lucas Segarra et al., 2019). These adjustments are capped at reasonable limits to ensure that the values remain within plausible ranges.

- **Physical-Model-based Solar Generation Data**: Using these adjusted meteorological features, we apply the *physical* model 4.1.1 to re-compute the solar generation data, reflecting potential real-world variations due to forecast errors. The estimated solar generation data is then combined with the ground truth net load data to update the disaggregated loads.

## 5 EXPERIMENTS

### 5.1 DATASETS

**Dataset for Net Load Forecasting**: We evaluate SONNET, the proposed probabilistic net load forecasting method, using the real-world dataset from the DOE net load forecasting competition (HeroX, 2023). This dataset comprises approximately one and a half years of hourly net load data, spanning from January 1, 2022, to July 16, 2023, from four city/town-size locations across different states—Texas (TX), Oregon (OR), Georgia (GA), and Hawaii (HI). These four distinct U.S. locations, situated in the South Central (TX), Northwest (OR), Southeast (GA), and Pacific Island (HI) regions, have very different weather patterns and solar penetration levels. This allows us to assess the generalizability of SONNET across diverse climatic conditions and renewable energy penetration levels. The competition period spanned from June 18, 2023, to July 15, 2023, during which the competing teams were required to submit their probabilistic day-ahead net load forecasts for the following day. For privacy considerations, the net load data is quantized and normalized relative to the peak net load at each location. Additional data provided includes approximate (again for privacy protection) latitude and longitude, as well as monthly solar capacity estimates $C$ for each location. The normalized solar capacity (normalized by the maximum net load) in TX, OR, GA, and HI are approximately 0.18, 0.35, 0.63, and 1.57, respectively, reflecting the diverse solar penetration levels. The very high solar penetration level of HI, as well as HI's uniquely poor day-ahead weather forecast accuracy, pose the greatest challenge for net load forecasting among the four.

**Dataset for Solar Disaggregation**: As an intermediate step, we would also like to assess the solar disaggregation component of our method. This is impossible to perform using the DOE competition dataset due to the absence of BTM solar generation data. We thus use an alternative dataset from Austin, TX, where ground truth solar generation information is available (Holcomb, 2012). This dataset comprises smart meter data from 322 customers over four weeks from August 3, 2015, to August 30, 2015. A subset of these customers is equipped with BTM solar systems. We apply our unsupervised solar disaggregation algorithm to the aggregate net load data from all the customers in the dataset, evaluating its effectiveness in a real-world setting.

### 5.2 SOLAR DISAGGREGATION

We begin with solar disaggregation with the settings above where approximate location is provided and net load data is normalized by the max net. Next, we train a supervised physical model based on the ground truth solar generation data and use the estimated capacity as the ground truth monthly solar capacity for this region. Since the amount of solar generation at any given time is proportional to the solar capacity according to equation 3, we scale the solar generation in this dataset to simulate different solar penetration levels, specifically, the same ones as the DOE competition sites, 0.18, 0.35, 0.63, and 1.57, respectively.

Table 1: Performance evaluation of the proposed unsupervised disaggregation algorithm.

| Capacity | RMSE | | MASE | | CV | |
|---|---|---|---|---|---|---|
| | Supervised | Unsupervised | Supervised | Unsupervised | Supervised | Unsupervised |
| 0.18 | 0.005 | 0.009 | 0.328 | 0.565 | 0.188 | 0.331 |
| 0.35 | 0.010 | 0.012 | 0.328 | 0.407 | 0.188 | 0.237 |
| 0.63 | 0.017 | 0.018 | 0.328 | 0.372 | 0.188 | 0.201 |
| 1.57 | 0.043 | 0.044 | 0.328 | 0.347 | 0.188 | 0.193 |

To evaluate the performance, we compare the results from our unsupervised disaggregation algorithm with the performance bounds obtained from supervised learning. The evaluation metrics are root mean square error (RMSE), mean absolute scaled error (MASE), and coefficient of variation (CV) (Pu and Zhao, 2023). Table 1 demonstrates that the unsupervised solar disaggregation algorithm achieves performance very close to the supervised one. Visualizations of the results are provided in Appendix B.1.

## 5.3 PROBABILISTIC NET LOAD FORECASTING

Next, we evaluate the performance of the main task of this work — day-ahead probabilistic forecasting of net loads by SONNET. The predictor input includes historical weather variables, weather forecasts, solar generation forecasts, time-related variables, as well as disaggregated historical solar generation and loads. Detailed information about the inputs is provided in Appendix A.7. We leave the inclusion of other features as future work.

**Simulating Weather Forecasts.** To evaluate the performance of the proposed method in practical settings and ensure a fair comparison with the competition results, we simulate weather forecast errors as follows: Errors are randomly sampled from a normal distribution $N(0, \sigma_f^2)$, where $\sigma_f$ represents the *empirical* standard deviation of a feature $f$'s forecast error that we observe for a given hour and location. These generated errors are then added back to the ground truth values. The empirical standard deviations used in generating these errors are provided in Appendix B.2. To increase the level of difficulty in the net load forecasting task, we further simulate two additional settings, "Challenging" and "Extreme", where the errors are generated with 1.5x and 2x of the empirical forecast error standard deviation of each weather feature.

**Evaluation Metric.** Continuous Ranked Probability Skill Score (CRPSS) is used as the evaultion metric, which is the same metric as employed by the DOE competition:

$$\text{CRPSS} = \left(1 - \frac{\text{CRPS}_{\text{model}}}{\text{CRPS}_{\text{ref}}}\right) \tag{14}$$

where $\text{CRPS}_{\text{model}}$ is the CRPS score of the employed model, and $\text{CRPS}_{\text{ref}}$ is the CRPS score of a "reference" model provided by the DOE competition. The reference model simply calculates the forecast by aggregating the net load on an hourly basis from the last 30 days and calculating the probability of net load values for every hour of the next day (Doubleday et al.). Following the standard of the DOE competition, to represent a CDF, 11 quantiles (i.e. $0_{th}, 10_{th}..100_{th}$) of the distribution are to be estimated for each hour's net load. A perfect forecast would lead to a maximum CRPSS of 1. A negative CRPSS implies that the employed model performs worse than the reference model.

**Results.** We evaluate SONNET in practical settings by considering weather forecast errors and comparing our results with those achieved by the top teams in the DOE competition (cf. Table 2). The results demonstrate that, even under the most challenging conditions with extreme forecast errors of weather features, SONNET still consistently outperforms these top teams (which, in comparison, perform with normal errors). We further observe that, across these four locations, the higher the solar penetration level, the harder it appears to forecast net loads.

In Figure 2, visualizations of the probabilistic forecasts are depicted: we observe that, even with very poor weather forecasts under the "extreme" error mode, the probabilistic forecasts of SONNET are still both much more accurate and less uncertain than the reference. More visualizations are available in Appendix B.3.

## 5.4 ABLATION STUDY

In Table 3, we conduct a number of ablation studies to demonstrate the importance of the innovative components of SONNET. We observe the following key messages.

**Training without Disaggregated Solar** First, even without employing the solar disaggregation step in the absence of solar capacity information, our transformer-based model significantly outperforms state-of-the-art methods. Decomposing net load into separate components of load and solar generation

Table 2: This table presents the CRPSS (higher scores indicate better performance) of different methods under "Normal", "Challenging", and "Extreme" weather forecasting scenarios. The CRPSS scores for SONNET are the averages from 20 experiments. In each experiment, the model is evaluated over 10 replicates of the 28-day competition period, each with independently and randomly generated weather forecast errors. "Best among all" denotes the highest CRPSS score achieved for each location among all participating teams in the competition. Additionally, the scores of the top six teams, ranked by their average CRPSS across the four locations, are listed (HeroX, 2023). As the solar capacity increases across the locations (from TX to HI), all the models' performance decreases.

| Team | TX | OR | GA | HI | Average | Mode |
|---|---|---|---|---|---|---|
| Best among all | 0.581 | 0.446 | 0.280 | 0.014 | 0.330 | / |
| Garnet | 0.581 | 0.313 | 0.185 | -0.041 | 0.260 | / |
| Pearl | 0.510 | 0.287 | 0.208 | 0.013 | 0.255 | / |
| Turquoise | 0.499 | 0.401 | 0.134 | -0.072 | 0.241 | / |
| Quartz | 0.432 | 0.288 | 0.228 | -0.039 | 0.227 | / |
| Chrysocolla | 0.307 | 0.310 | 0.280 | -0.060 | 0.209 | / |
| Cuprite | 0.486 | 0.416 | 0.092 | -0.060 | 0.202 | / |
| SONNET | **0.628 ± 0.032** | **0.535 ± 0.024** | **0.293 ± 0.029** | **0.180 ± 0.016** | **0.409** | Normal |
| SONNET | 0.615 ± 0.025 | 0.517 ± 0.024 | 0.265 ± 0.026 | 0.104 ± 0.019 | 0.375 | Challenging |
| SONNET | 0.595 ± 0.019 | 0.493 ± 0.024 | 0.230 ± 0.024 | 0.019 ± 0.023 | 0.334 | Extreme |

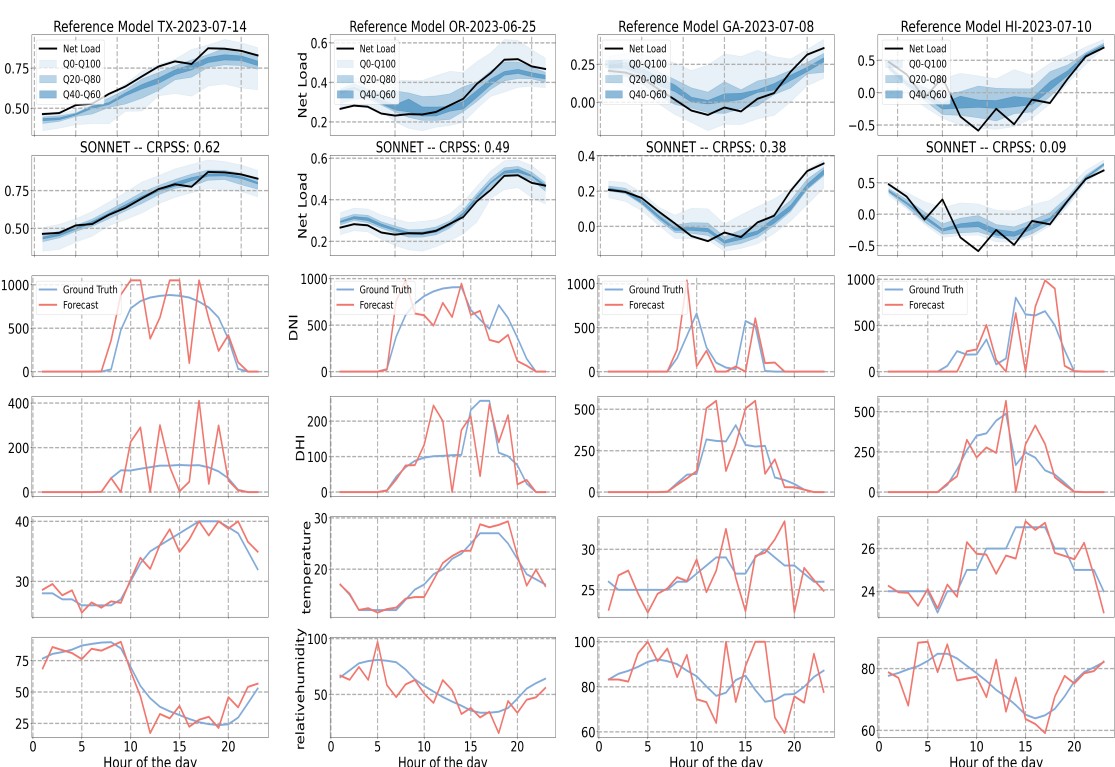

Figure 2: Net load forecasting for four locations under the "extreme" weather forecast error mode.

further enhances the model's ability to capture relationships between weather variables, time variables, and net load. This decomposition clarifies these relationships, resulting in improved forecasting accuracy.

**Training without Data Augmentation** If the model is trained solely on ground truth net load and weather data, it lacks robustness to weather forecast errors, especially in regions with high solar penetration levels such as HI. After applying data augmentation, the performance of all models improves significantly.

Table 3: Ablation Study

|  | TX | OR | GA | HI | Average |
|---|---|---|---|---|---|
| SONNET | **0.628** | **0.535** | **0.293** | **0.180** | **0.409** |
| w/o disaggregation | 0.621 | 0.528 | 0.292 | 0.167 | 0.402 |
| w/o data augmentation | 0.529 | 0.492 | 0.210 | 0.01 | 0.311 |
| w/o weather, w/o disaggregation | -0.356 | 0.043 | 0.100 | -0.015 | -0.057 |

**Training without Exogenous Variables**   In Table 3, we conduct experiments where no exogenous variables (weather) or solar disaggregation are included in the training and testing. These experiments demonstrate that time-series models relying solely on historical net load data suffer from much poorer performance.

**Training with Alternative Predictor Models**   In Table 4, We conduct experiments to compare SONNET with additional baselines based on different predictor models: a) We replace the encoder part of the Transformer with an LSTM and the decoder part with an MLP; b) We employ MLP followed by a cross attention mechanism; c) We employ XGBoostLSS as opposed to neural networks (März, 2019). We observe that SONNET significantly outperforms baseline methods across all locations.

Table 4: Performance Comparison with Other Models

|  | TX | OR | GA | HI | Average |
|---|---|---|---|---|---|
| SONNET (ours) | **0.628 ± 0.032** | **0.535 ± 0.024** | **0.293 ± 0.029** | **0.180 ± 0.016** | **0.409** |
| LSTM | 0.367 ± 0.104 | 0.414 ± 0.025 | 0.199 ± 0.045 | 0.074 ± 0.034 | 0.264 |
| XGBoostLSS | 0.406 ± 0.0198 | 0.157 ± 0.01 | 0.091 ± 0.023 | 0.055 ± 0.040 | 0.178 |
| MLP | 0.348 ± 0.110 | 0.274 ± 0.130 | 0.113 ± 0.055 | -0.109 ± 0.076 | 0.157 |

**Training with Different Context Lengths**   In Table 5, we conduct an ablation study with different context lengths, i.e., the lengths of the historical look-back window utilized in the Transformer model. We observe that the context length of 14 days (our default) achieves the best performance, while 7 and 21 days are slightly worse.

Table 5: Ablation Study with Different Look-Back Context Lengths

|  | TX | OR | GA | HI | Average |
|---|---|---|---|---|---|
| SONNET (context window = 7 days) | 0.628 | 0.517 | 0.299 | 0.180 | 0.406 |
| SONNET (context window = 14 days) | **0.628** | **0.535** | **0.293** | **0.180** | **0.409** |
| SONNET (context window = 21 days) | 0.562 | 0.555 | 0.266 | 0.179 | 0.390 |

## 6   CONCLUSION

We developed SONNET, a novel method for net load forecasting based on disaggregated BTM solar generation and loads. An enhanced Transformer architecture that takes both historical data and exogenous future input such as weather forecast is designed. A physical-model-based data augmentation technique is developed to improve the predictor robustness to weather forecast errors. Extensive validation based on real-world data from U.S. DOE's recent net load forecasting competition demonstrated that our method consistently and significantly outperforms the state of the art. Last but not least, the developed techniques, in particular, disaggregation, enhanced Transformer architecture, and physics model-based data augmentation, all have broad applications beyond net load forecasting.

ETHICS STATEMENT

Our study is solely focused on addressing the scientific problem, and therefore does not present any foreseeable ethical concerns or implications.

REPRODUCIBILITY STATEMENT

In the main text, we have rigorously formalized our model framework and task settings with mathematical formulas. We include the implementation details in Appendix. The source code is provided in supplementary materials will be made public once the paper is accepted.

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

# Appendix

## A    EXPERIMENTAL DETAILS

In this section, we present details on solar disaggregation, data augmentation, and training the probabilistic net load forecasting model.

### A.1    WEATHER DATASET

We download the hourly weather data including, Direct Normal Irradiance (DNI), Diffuse Horizontal Irradiance (DHI), temperature, relative humidity, and zenith from Solcast (Solcast, 2019) with the given latitude and longitude for each location.

### A.2    SOLAR DISAGGREGATION

#### A.2.1    PHYSICAL MODELS

In equation 4, $\alpha$, $\tau_{b,t}$ and $\rho_t$ are the elevation of the sun, atmospheric transparent coefficient and surface albedo, respectively. $\alpha$ can be calculated by equation 17. The cell temperature $T_{PV,t}$ is approximated by:

$$T_{PV,t} = T_{A,t} + \frac{I_{PV,t}}{800} \times (N_{oct} - 20), \tag{15}$$

where $T_{A,t}$ is the ambient temperature and $N_{oct}$ is the Nominal Operating Cell Temperature. $I_{b,t}$ is the direct horizontal irradiance which can be calculated as follows

$$I_{b,t} = I_{0,t}\tau_{b,t} \sin \alpha \tag{16}$$

$$\alpha = 90 - \zeta, \tag{17}$$

where $\zeta$ is the zenith of the sun. $A$ in equation 4 denotes the azimuth of the sun and can be calculated as the following:

$$\cos A = \frac{\sin \delta \cos \phi - \cos \delta \sin \phi \cos \omega}{\cos \alpha}, \tag{18}$$

$$\delta = 2\pi \times \frac{23.45°}{360°} \times \sin\left(2\pi \times \frac{284 + N}{Y}\right). \tag{19}$$

The above equation only gives the correct estimation of azimuth in the solar morning(i.e. $\omega < 0$), so post calculation correction needs to be applied(Wikipedia contributors, 2023):

$$A = \begin{cases} A & if \omega < 0 \\ 360 - A & if \omega >= 0, \end{cases} \tag{20}$$

Where $\omega$ denotes the hour angle which is negative before 12:00 and $D$ is the time zone of the given location.

$$\omega = (t - 12) \times 15° + (\psi - D \times 15°). \tag{21}$$

In the disaggregation, $\mu$ denotes the temperature coefficient, with $-0.5\%/°C$ as a typical value. $I_{ref}$ and $T_{ref}$ are the reference irradiance and cell temperature, set to be $1000 \text{ W/m}^2$ and 25, respectively. $N_{oct}$ is the Nominal Operating Cell Temperature, with 48 as a typical value (Dong and David, 2017). $\tau_{b,t}$ and $\rho_t$ are set to be 0.74 and 0.2, respectively (Dobos, 2014).

#### A.2.2    INITIALIZATION OF TRAINING SET PAIRS

We initialize the training set as follows: for all neighboring time pairs within the daytime, we collect all the time pairs $(t, t')$ which satisfy $(N_t - N_{t'}) \cdot (I_{PV,t} - I_{PV,t'}) < 0$. This condition ensures that for each time pair, the higher net load corresponds to lower received irradiance, increasing the likelihood that the loads are similar. Among these pairs, we select the ones with the largest difference in terms of the solar irradiance received on the solar panels ($I_{PV,t}$).

### A.2.3 UNSUPERVISED SOLAR DISAGGREGATION

During the training, we set $M$ to be 100 meaning that we collect 100 time pairs in each iteration. $\gamma$ and $\eta$ are both set to 1, and the maximum number of iterations in the disaggregation algorithm is set to be 10.

## A.3 POSITION EMBEDDING

Suppose the input tensor $\boldsymbol{x} \in \mathbb{R}^{d_f \times d_t}$ has a feature dimension $d_f$ and temporal dimension $d_t$. The input tokens are mapped to the Transformer latent space of dimension $d$ via a trainable linear projection $\mathbf{W}_p \in \mathbb{R}^{d \times d_f}$, and a learnable additive position encoding $\mathbf{W}_{\text{pos}} \in \mathbb{R}^{d \times d_t}$ is applied to monitor the temporal order of patches: $\boldsymbol{x}' = \text{PE}(\boldsymbol{x}) = \mathbf{W}_p \boldsymbol{x} + \mathbf{W}_{\text{pos}}$, where $\boldsymbol{x}' \in \mathbb{R}^{d \times d_t}$ denote the input that will be fed into Transformer encoder.

## A.4 MULTI-HEAD ATTENTION

We employ a standard Transformer encoder with multi-head attention (MHA) to map the observed signals to latent representations. The MHA block can be represented as $\boldsymbol{z} = \text{MHA}(\boldsymbol{x}^Q, \boldsymbol{x}^K, \boldsymbol{x}^V)$, where  is the output of MHA, and $\boldsymbol{x}^Q, \boldsymbol{x}^K, \boldsymbol{x}^V$ are the input for query, key and value respectively.

Each head $h = 1, ..., H$ in MHA will transform the input into query matrices $Q_h = (\boldsymbol{x}^Q)^T \mathbf{W}_h^Q$, key matrices $K_h = (\boldsymbol{x}^K)^T \mathbf{W}_h^K$ and value matrices $V_h = (\boldsymbol{x}^V)^T \mathbf{W}_h^V$, where $\mathbf{W}_h^Q, \mathbf{W}_h^K, \mathbf{W}_h^V$ are learnable parameters. After that a scaled production is used for getting attention output $\boldsymbol{z}_h$:

$$\boldsymbol{z}_h^T = \text{Attention}(Q_h, K_h, V_h) = \text{Softmax}(\frac{Q_h K_h{}^T}{\sqrt{d_k}})V_h$$

where $d_k$ is the dimension of the key vectors. Then all the output from different heads will be concatenated. The MHA block also includes BatchNorm, which is preferred in time series Transformer as demonstrated in (Zerveas et al., 2021), and a feed forward network with residual connections to generate the final output $\boldsymbol{z}$, which is similar to the design in vanilla Transformer (Vaswani et al., 2017).

## A.5 CRPS LOSS

The CRPS loss is not directly available in standard PyTorch libraries. We implement a CRPS loss for general usage and our experiments, which is as follows:

```python
class CRPSLoss(nn.Module):
    def __init__(self,
                 quantiles=[0,0.1,0.2,0.3,0.4,0.5,0.6,0.7,0.8,0.9,1],
                 adjusted=True,
                 eps=1e-10):
        super().__init__()
        self.adjusted = adjusted
        if self.adjusted:
            self.quantiles = torch.tensor([0]+quantiles+[1])
        else:
            self.quantiles = torch.tensor(quantiles)
        self.eps = eps

    def forward(self, preds, target):
        assert not target.requires_grad
        assert preds.size(0) == target.size(0)
        if self.adjusted:
            B = preds.shape[0]
            N = preds.shape[1]
            T = preds.shape[2]
```

```
              max_bound = 100
              min_bound = -100
              preds = torch.cat((min_bound*torch.ones((B, N, T, 1),
                                          device=preds.device),
                                     preds),
                                     dim=-1)
              preds = torch.cat((preds,
                                     max_bound*torch.ones((B, N, T, 1),
                                     device=preds.device)),
                                     dim=-1)

          q_i1 = self.quantiles[1:].to(preds.device)
          q_i = self.quantiles[:-1].to(preds.device)
          X_i1 = preds[:,:,:,1:]
          X_i = preds[:,:,:,:-1]
          X_t = target.unsqueeze(3).repeat(1, 1, 1, X_i.shape[-1])

          index = torch.full_like(X_i1, 2)
          index[X_t > X_i1] = 0
          index[X_t < X_i] = 1
          index = F.one_hot(index.to(torch.int64), num_classes=3)

          term0 = 1/3*torch.einsum('bntq,q->bntq',
                                      X_i1-X_i,
                                      (q_i1**2+q_i1*q_i+q_i**2))
          term1 = 1/3*torch.einsum('bntq,q->bntq',
                                      X_i1-X_i,
                                      ((q_i1-1)**2+(q_i1-1)*(q_i-1)
                                      +(q_i-1)**2))
          term2 = torch.einsum('bntq,q->bntq',
                                      X_t-X_i, 2*q_i-1) +
                                      torch.einsum('bntq,q->bntq',
                                      (X_t-X_i)**2/(X_i1-X_i+self.eps),
                                      q_i1-q_i) + term1
          terms = torch.stack((term0,term1,term2),dim=-1)

          loss = torch.einsum('bntqd,bntqd->bntq',
          index.to(torch.float), terms)
          return torch.mean(torch.sum(loss,dim=-1))
```

## A.6 DATA AUGMENTATION

The training dataset comprises one copy of the ground truth data and two copies of the augmented data. For the augmented data, we draw the error ratio term $\epsilon$ drawn from a normal distribution: $N(0, 0.36)$ for DNI/DHI and draw the error term from $N(0, 2)$ and $N(0, 12)$ for temperature and relative humidity, respectively (Lucas Segarra et al., 2019). To ensure that the synthetic weather variables remain within plausible ranges, we clip the generated DNI/DHI values to fall between zero and the maximum value observed in the ground truth data. Similarly, the relative humidity values are clipped to lie within the range of 0 to 100.

## A.7 TRAINING PROBABILISTIC NET LOAD FORECASTING MODEL

### A.7.1 INPUT VARIABLES

Formally, the net load variables $X(t)$ and the exogenous variables $S(t)$ (which includes $W(t)$ and $Z(t)$) on a given time step $t$ can be represented as:

$$X(t) \subseteq \{\text{net load}(t), \text{load}(t), \text{disaggregated solar generation}(t)\},$$
$$S(t) = \{\text{temperature}(t), \text{humidity}(t), \text{DHI}(t), \text{DNI}(t), \text{time features}(t)\}$$

The exogenous variables could provide extra information beside historical net load data, which is beneficial to the forecasting (Wang et al., 2024; 2023a). The inputs for SONNET (and methods in the ablation study) are shown in Table 6. Time-related variables are encoded using cyclical encoding since cyclical patterns are important in time series data (Chen et al., 2022; Wang et al., 2022). For a time feature $z$, its embedding can be expressed as:

$$\left[ \sin\left( \frac{2\pi z}{\omega(z)} \right), \cos\left( \frac{2\pi z}{\omega(z)} \right) \right] \tag{22}$$

Where $\omega(z)$ is the frequency for time feature $z$.

Table 6: $W$ includes weather features such as DNI, DHI, temperature, and relative humidity. $Z$ includes time-related features such as year, dayofyear, month, weekday, and hour. Weather features in the future input are generated as described in Sec 5.3. Load is the summation of net load and disaggregated solar.

| Method | Historical Input | Future Input |
|---|---|---|
| SONNET | Load + Disaggregated solar + $W$ + $Z$ | Forecast $W$ & solar + $Z$ |
| W/O disaggregation | Net load + $W$ + $Z$ | Forecast $W$ + $Z$ |
| W/O weather, W/O disaggregation | Net load + $Z$ | $Z$ |

### A.7.2    IMPLEMENTATION DETAILS AND COMPUTATIONAL EFFICIENCY.

The data prior to May 1st, 2023 is used for training; the data from May 1st to June 15th, 2023 is used for validation; the test set is from June 18th, 2023 to July 16th, 2023, which is the same as the competition. The model is trained for up to 300 epochs with an early stopping patience of 30 epochs. If early stopping is triggered at epoch $x$, the model is re-trained using both the training and validation datasets for $x$ epochs. This approach ensures the model fully leverages the limited available data. We employs 8 attention heads for all locations. For the "HI" and "OR" locations, the embedding dimension ($d$) is set to 64. However, for the "GA" location, $d$ is reduced to 48. This is because the data from "GA" is heavily quantized and approximately one-third of it is dropped due to quality issues. The model architecture includes a single Transformer encoder layer combined with a cross attention layer with a dropout rate of 0.3. The look back window $l$ is 336 (hours), and the historical input data is segmented into patches with a length of 8 and a stride of 4, while the prediction window is 24 (hours) and no patching is applied to the future input data. Learning rate in all experiments are automatically selected by employing the same algorithm as in (Nie et al., 2022). The model is trained on a single RTX 3090 GPU and takes approximately 10-20 minutes to complete training for one location, using augmented data spanning roughly four and a half years.

## B    ADDITIONAL RESULTS AND VISUALISATIONS

### B.1    SOLAR DISAGGREGATION UNDER DIFFERENT CAPACITY

This section presents the performance of BTM solar generation estimation under different solar capacity, see Figure 3.

### B.2    WEATHER FORECAST ERROR GENERATION

This section presents the empirical hourly standard deviation, see Figure 4.

### B.3    MORE VISUALIZATIONS FOR NET LOAD FORECASTING

More visualizations are provided below, see Figures 5, 6, 7,and 8.

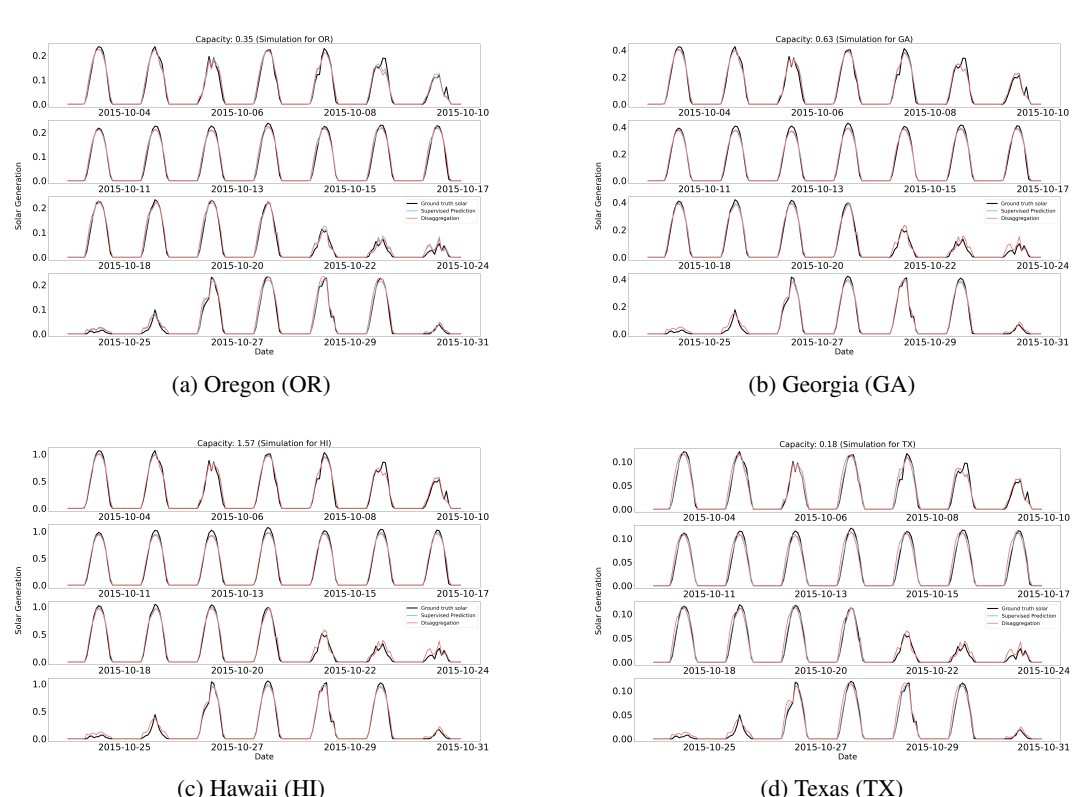

Figure 3: Solar Disaggregation Results for Different Locations

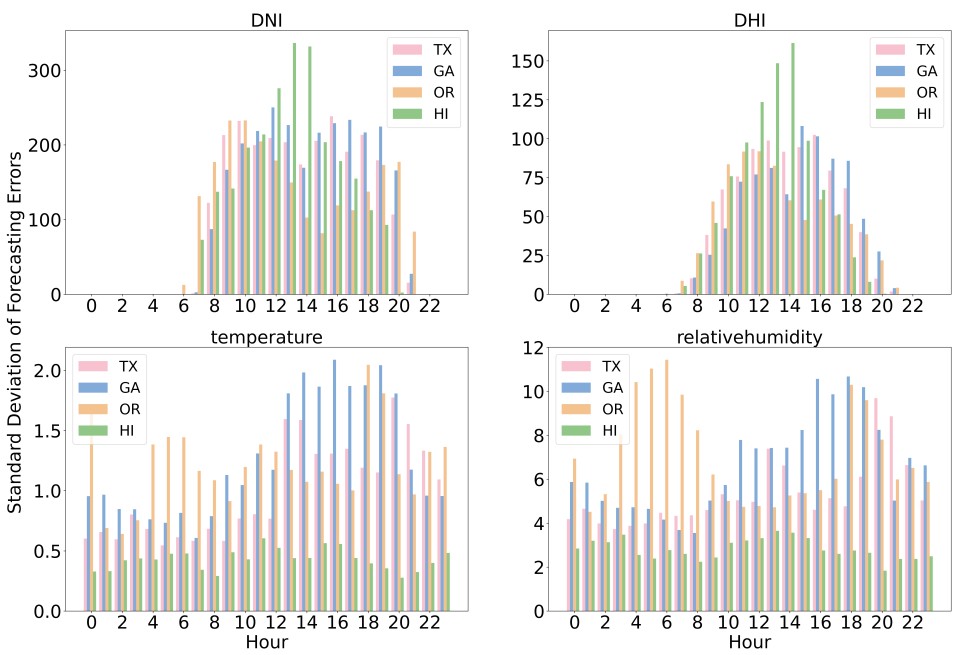

Figure 4: Empirical hourly standard deviation of forecast errors for different weather variables

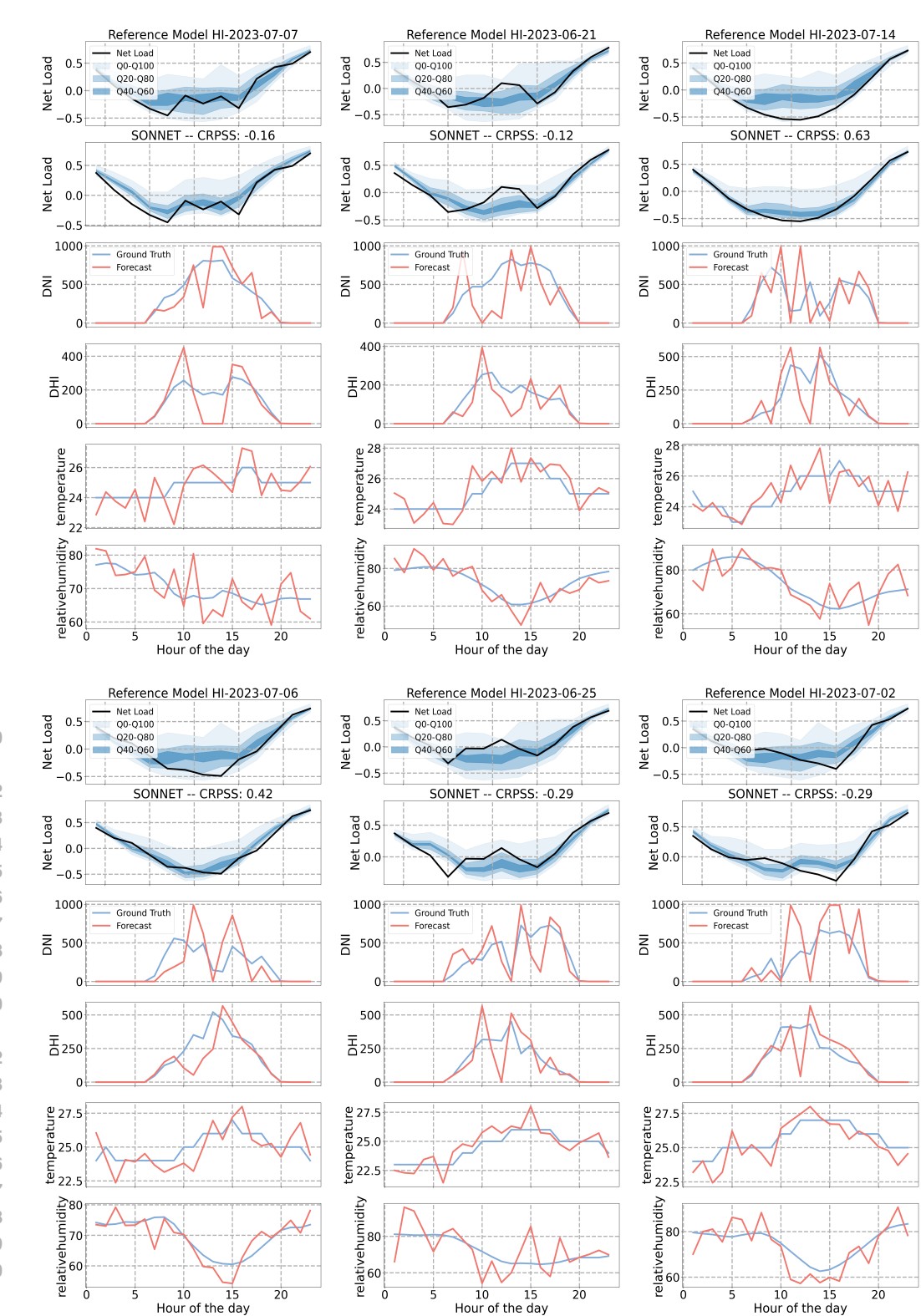

Figure 5: Net load forecasting for HI under "extreme" weather forecast error mode for a subset of 6 randomly selected days.

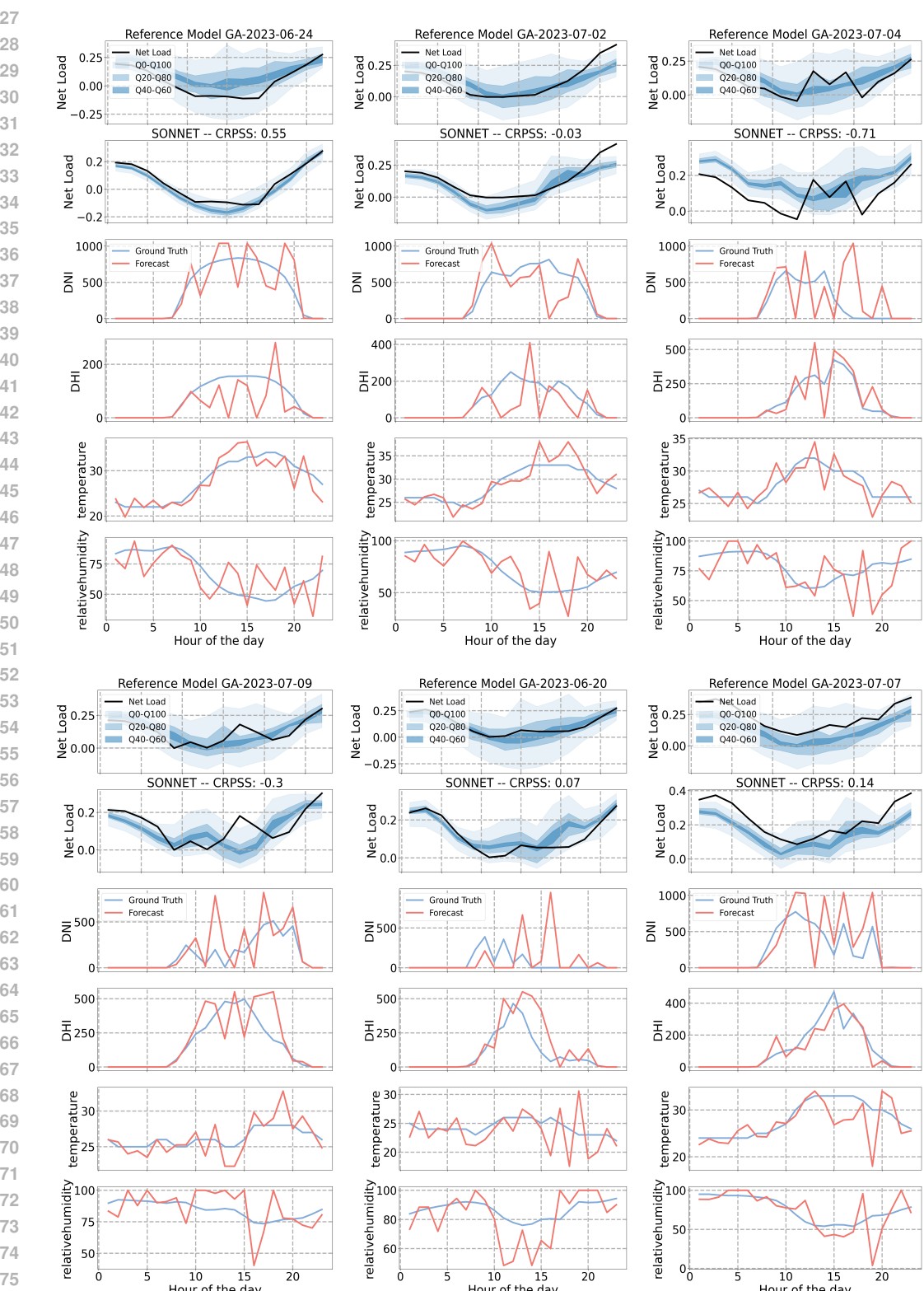

Figure 6: Net load forecasting for GA under "extreme" weather forecast error mode for a subset of 6 randomly selected days.

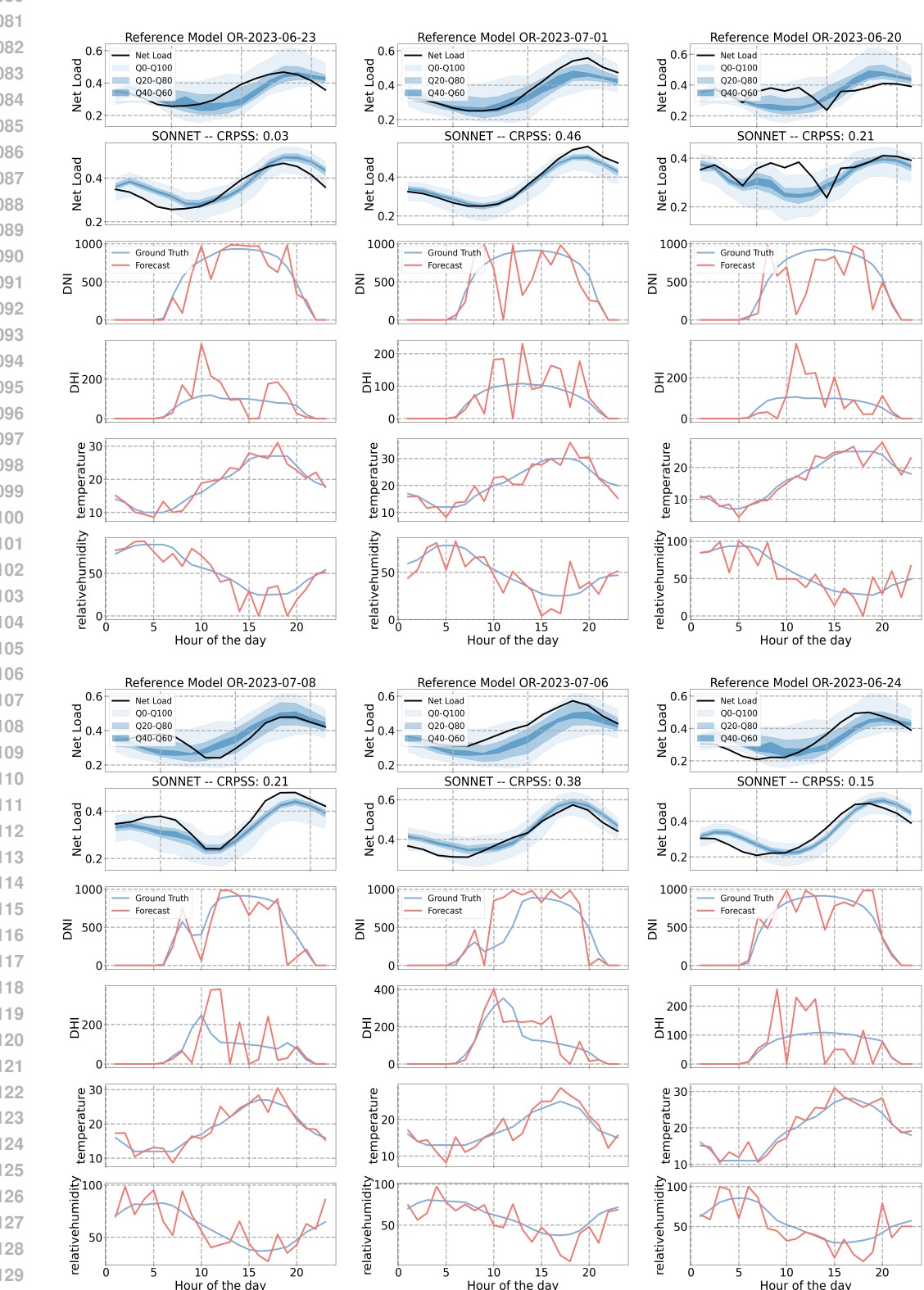

Figure 7: Net load forecasting for OR under "extreme" weather forecast error mode for a subset of 6 randomly selected days.

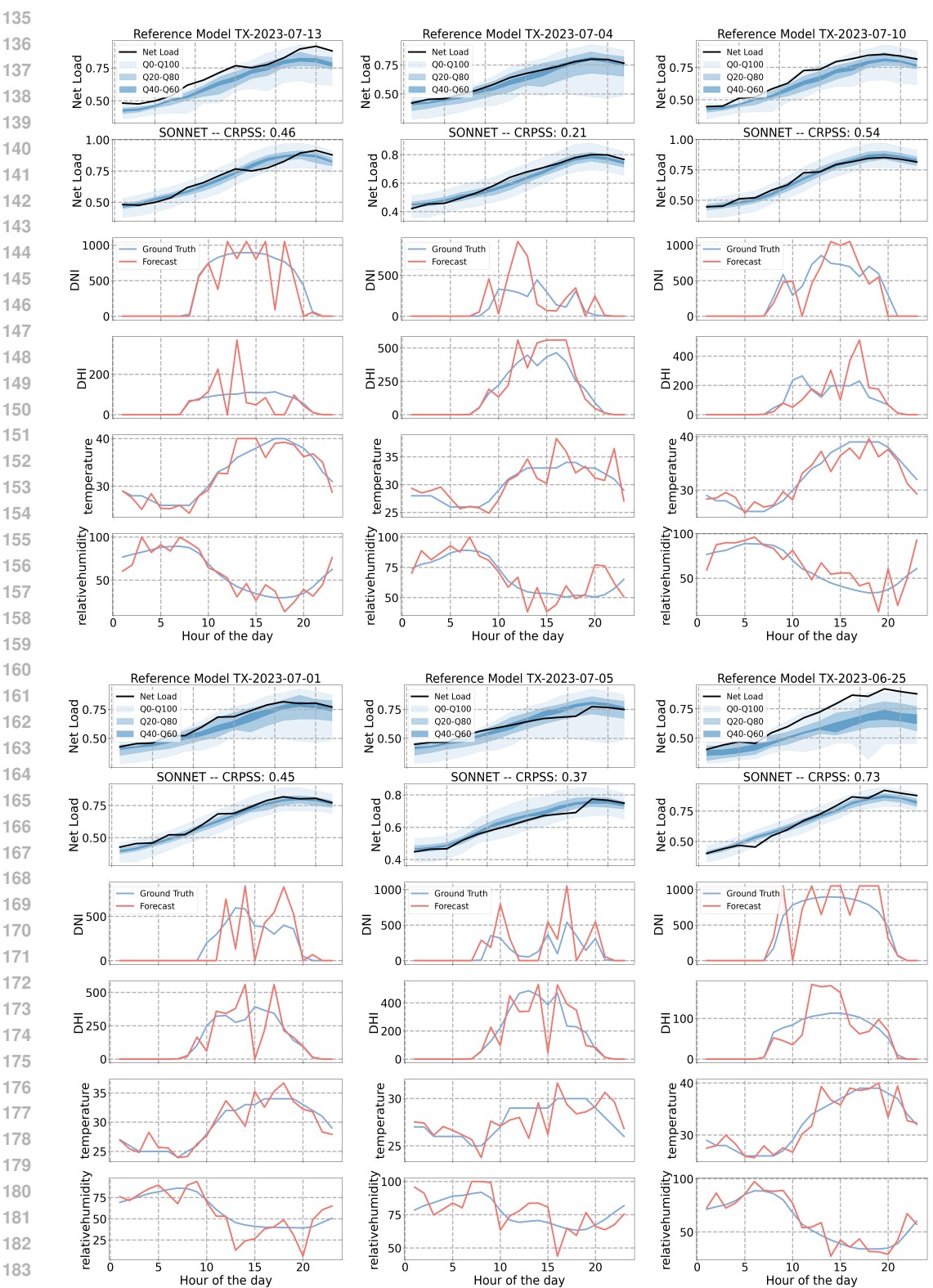

Figure 8: Net load forecasting for TX under "extreme" weather forecast error mode for a subset of 6 randomly selected days.

