# OpenReview forum: "SONNET: Solar-disaggregation-based Day-ahead Probabilistic Net Load Forecasting with Transformers"
_ICLR.cc/2025/Conference — ICLR 2025 Conference Withdrawn Submission_

### Official Review · Reviewer_ZZxR · 2024-10-22

**Soundness:** 3
**Presentation:** 3
**Contribution:** 2
**Rating:** 5
**Confidence:** 3

**Summary:**

This paper shows the new forecasting problem-net load forecasting. Then, the paper proposed SOONET. The SONNET method is used for day-ahead probabilistic net load forecasting. SOONET includes an unsupervised BTM solar decomposition algorithm, a transformer-based architecture that can deal with historical and exogenous data, and a physical model-based approach to data enhancement is also used in the way.

**Strengths:**

This paper contains unsupervised solar disaggregation, a Transformer enhanced by a physical model. An effective data augmentation for robustness and limited solar data.SOONET also conducts a real datasets to verify the model.

**Weaknesses:**

I think this article lacks novelty. For example, the model architecture is very similar to that of TimeXer.The author is to transform the existing method into a practical scenario.At the same time, solar power generation has a great influence on weather, and the author seems to have not fully considered the influence of weather images on solar power generation prediction.I think the authors need to conduct more extensive benchmark experiments to prove the effectiveness of the proposed method

**Questions:**

1. Why does the author use the Transformer-based framework and not compare individual models (such as MLP-based)?
2. The author only compared the baseline related to the competition, so why not add the comparison with other baselines (Autoformer or Dlinear)?
3. From the perspective of the model, the framework proposed by the author seems to be very similar to TimeXer. Please point out the difference between the framework and the TimeXer(TimeXer: Empowering Transformers for Time Series Forecasting with Exogenous Variables)

---

### Official Review · Reviewer_Ghm5 · 2024-10-25

**Soundness:** 1
**Presentation:** 3
**Contribution:** 2
**Rating:** 3
**Confidence:** 5

**Summary:**

This study propose a novel probabilistic net load forecasting method based on disaggregating net loads into solar generation and loads and feeding both into the predictors. The model developed based on an enhanced Transformer architecture that integrates both historical and
future input data, employing a combination of self-attention and cross-attention mechanisms, and a data augmentation method that enhances the robustness of net load forecasts against weather forecast errors.

**Strengths:**

The paper is logically structured and clearly presented, allowing readers to follow and comprehend the content with ease.

**Weaknesses:**

The paper’s shortcomings include an incomplete literature review, the use of inappropriate metrics and insufficient mathematical descriptions, limited contribution, the absence of SOTA comparisons, and a lack of necessary statistical methods. Please see more details in Questions.

**Questions:**

1, Net Load Forecasting, particularly in probabilistic forms, is an active research area in the energy field. However, this study lacks an adequate literature review to clearly identify the research gap between existing studies and the proposed work. Authors can find recent reseach on several reputable journals in the field of energy, for example Applied Energy, Energy and IEEE Transactions on Power Systems

2, In the validation process, only the Continuous Ranked Probability Skill Score (CRPSS) is used to assess the model’s performance. However, for a probabilistic forecasting model, additional evaluations are needed, such as point-wise metrics (e.g., RMSE, MAE) and distribution similarity measures (e.g., Wasserstein Distance), which are missing from this study. Please explain the rationale for focusing solely on CRPSS, and please add more metrics in revision version.

3, The study lacks necessary statistical tests to confirm that the observed performance improvements are not due to chance. For example, in Tables 3 and 5, the performance differences between models are minimal, raising the question of whether these differences are significant enough to support the conclusions drawn. Please conduct Diebold-Mariano Test to validate the results.

4, The study does not include sufficient comparisons with state-of-the-art (SOTA) models. Although Tables 2 and 4 provide comparisons with some competing models and basic benchmarks, it does not test the performance against mature and up-to-date probabilistic Net Load Forecasting models. Please add  2~3 SOTA models which published after year 2023 in comparsion test.

5, In the disaggregation process, the authors note that the DOE competition dataset could not be used due to the absence of behind-the-meter (BTM) solar generation data, leading to the use of Austin data as an alternative. However, this approach is problematic because solar forecasting is highly sensitive to time and geographic location, introducing potential bias and errors. Please discuss the potential limitations or biases introduced by using the Austin dataset for disaggregation and how author address these limitations.

**Details Of Ethics Concerns:**

Nan

---

### Official Review · Reviewer_5YSC · 2024-10-29

**Soundness:** 3
**Presentation:** 3
**Contribution:** 2
**Rating:** 5
**Confidence:** 3

**Summary:**

The paper proposes an approach for day-ahead probabilistic net load forecasting using transformers. The approach relies on a disaggregation mechanism that is also proposed in this work which utilizes weather predictions to disaggregate data into solar generation and remaining loads. The evaluation of the approach is based on the basic principles followed by the recent net load forecasting competition organized by the U.S. Department of Energy (DOE).

**Strengths:**

The paper tackles a timely and important issue. It introduces a complete methodology including a novel Similarity-based BTM Solar Disaggregation algorithm and a Transformer-based architecture designed for net load prediction. The paper is well writen, well organized and easy to follow.

**Weaknesses:**

Forecast errors are rarely independent and identically distributed (iid). Errors tend to cluster—e.g., an incorrect overcast prediction often leads to multiple subsequent incorrect forecasts throughout the day. This raises concerns about evaluating the model using errors sampled from a Gaussian distribution, as this may artificially inflate robustness compared to real-world scenarios. To substantiate robustness claims against weather forecast errors, the authors could consider testing the model using real weather forecasting data to capture the correlated nature of forecast inaccuracies. Alternatively, they could generate more realistic synthetic forecast errors by introducing temporal correlations, using, for instance, autoregressive models that reflect the clustering nature of forecast errors. This would provide a more comprehensive evaluation of the model’s performance under realistic conditions.

One of the biggest challenges in this domain is generating solar radiation predictions, as these can be transformed into PV power output predictions. The paper assumes solar radiation predictions are provided and treats them as inputs to the proposed approach. However, it's unclear whether the benchmark approaches also rely on such predictions as inputs, which raises concerns about the comparability and fairness of the results. I recommend the authors provide a detailed comparison, either in table format or as a section in the discussion, specifying the exact inputs used by each benchmark method. Highlighting potential differences will help clarify whether the comparison between the proposed approach and the benchmarks is fair and meaningful.

The disaggregation mechanism presented in the paper could be strengthened by comparing it with alternative approaches. For example, the authors could compare their method to one that estimates the theta parameters based on the optimal angles for the specific latitude and longitude, combined with a simple multiplicative correction factor. This comparison would provide deeper insights into the relative strengths and weaknesses of the disaggregation method. If this comparison is beyond the scope of the current paper, it would be helpful for the authors to explain why they chose their current approach over simpler alternatives and perhaps include a plan to explore these alternatives in future work.

**Questions:**

How the disaggregation model currently accounts for or could account for solar tracking?

---

### Official Review · Reviewer_JwrW · 2024-10-29

**Soundness:** 2
**Presentation:** 2
**Contribution:** 1
**Rating:** 3
**Confidence:** 4

**Summary:**

The paper presents a novel approach to solar load forecasting, making notable improvements in this specific domain. However, the general applicability of the methods beyond solar forecasting is unclear, primarily because the key contribution—a physics-inspired data augmentation technique—is closely tied to the solar load task.

Overall, the paper provides valuable advances for solar load forecasting, but its contributions may be less impactful in broader contexts without further evidence of generalizability. The dataset choice is quite narrow for a machine learning (ML) conference, and the paper might be more suited to a forecasting or renewable energy journal.

**Strengths:**

- Architecture: The authors introduce a sophisticated patch-based transformer architecture, employing both self-attention and cross-attention mechanisms to handle future exogenous features. This structure is well-suited for capturing complex temporal dependencies in solar load data.

- Performance Gains: Tested on the Herox solar load forecasting competition dataset, the method shows impressive improvements, achieving a 21% boost in the CRPSS metric over the state-of-the-art. This substantial leap in performance demonstrates the efficacy of the proposed model within this specific task.

**Weaknesses:**

- Narrow Focus: While the paper offers strong contributions to solar load forecasting, its generalizability to other domains remains questionable. The physics-based augmentation technique is heavily tailored to the nuances of solar data, making it difficult to assess whether the approach can be effectively applied to other forecasting tasks with different characteristics.

- Ablation Studies: The authors provide some ablation studies to isolate the effects of their proposed data augmentation and disaggregation techniques. Their findings indicate that the physics-based data augmentation is the primary driver of performance gains in their SONNET model, highlighting its importance in the overall method.

**Questions:**

1. The comparison in Table 4 is limited to LSTM, XGBoost, and MLP, ignoring many recent advances in neural forecasting. This narrow baseline selection weakens the paper's claims.

2. How does the SONNET method compare to simple baselines like ARIMA/ETS/SeasonalNaive?

3. To strengthen the analysis, it would be helpful to apply the data augmentation technique to the LSTM, XGBoost, and MLP models to assess its impact on those methods.

4. The comparison to Herox’s top performers in Table 2 is unclear, as the authors had access to the test set. If ablation studies were performed on the test set, it raises concerns about fairness.

5. The data augmentation technique appears overly simplistic, resembling a Kaggle competition trick (a normal multiplicative error) rather than a substantial ML contribution.

6. Given the $600,000 prize in the Herox competition, why didn’t the authors participate? Understanding the differences between the competition and this submission that led to a 21% improvement would clarify the novelty of the contribution.

---

### Official Review · Reviewer_7jS6 · 2024-11-03

**Soundness:** 1
**Presentation:** 3
**Contribution:** 3
**Rating:** 3
**Confidence:** 5

**Summary:**

This paper proposes an approach to improve day-ahead residual load predictions at high spatial resolution by disaggregating net load into electric consumption and behind-the-meter (BTM) generation from solar energy sources. The authors introduce a specialized Transformer model with data augmentation techniques aimed at enhancing prediction accuracy. They assert that disaggregating solar generation from the total load improves prediction performance by isolating solar-specific variabilities and patterns, potentially resulting in more accurate forecasting. The proposed method demonstrates improved accuracy in a case study, positioning this approach as relevant for renewable power system dispatch, particularly at granular levels, such as single residential, commercial and industrial buildings.

Abstract, Intro, Related work:
- Consider the use of "residual load" in addition to, or instead of, net load.
- Maybe worth mentioning: distribution systems are often operated as blackboxes. The most fine-grained mesaurements we can make is net load where smart meters are deployed in a power system.
- Maybe worth mentioning: despite being much more variable than load, solar irradiance also has a clear daily and seasonal pattern, similar to load, that can be predicted.
- Write "we demonstrate.." (first person) wherever you do something, and "it is demonstrated.." (third person) only where others have done something. This makes it more clear for the reader to figure out your concrete workings.
- Maybe worth mentioning: despite being in much smaller amounts prevalent, micro-wind may be relevant to mention in the context of BTM generation.
- Question on motivation: Why does a system operator need to know net load forecasts at the scale of single buildings, and not just at the aggregated scale of lower/medium voltage transformers only? This is a prediction that is much easier to make and suffices for planning and dispatch.
- It does not become clear to me, why forecasting models for load need to be different for forecasting net load.
- Why do you have to disaggregate load and solar generation first before making predictions of net load? Why don't you predict net load with the same information, which you used for disaggregation, right away? Need experiments to support this choice!
- You can feed in your predictions to the transformer right away, in a consistent/standardized manner, for example by using a dedicated and unique token as delimiter for the input between predictions and historic data. Why is a new architecture necessary? Need experiments to support this choice!
- Data augmentation competes with encoding the physics into architecture of transformer model, e.g. Physics informed Neural Networks (PINNs), reducing the problem volume space and requiring less data for training. What motivates your choice as opposed to this, and therefore makes your approach a contribution?
- It is true that day-ahead (net) load forecasting is an important task for power system dispatch (and planning to some extent). Again, what motivates the granularity of this needing to be at the scale of single residential buildings for instance and not at a much easier, aggregated level like low/medium voltage transformer substations.

Problem Formulation:
- It would be great to also formalize what are the inputs and what are the outputs to your model, including their dimensions, strucutre (sequences of what tensor dimension?).

Methodology:
- From a information theoretical perspective, what information do you add to net load forecasting through a disaggregation step, as opposed to processing the information used for disaggregation for the forecast right away?

Experiments:
- You bury the lede. It would be great to state results right away in this section.

Experiments - Ablation study:
- Training without Disaggregated Solar: You correctly mention that without disaggregation there is an absence of solar capacity information (and any other information used in the disaggregation step). Then, you show that the performance of your transformer model is still good. This raises the question of whether the disaggregation step is unnecessary, and that feeding all relevant information right away into the transformer is actually sufficient. A set of experiments that evaluates this is necessary but missing.
- Regardless of whether these experiments have been done or not, what is your intuition behind why introducing a disaggregation needs to increase performance? It seems that you are arguing with a disentanglement of important relationships for net load in the raw feature space, a disussion related to equivariant feature representation learning, but one that is not sufficiently understood or evaluated in the study.
- Missing: an evaluation of encoding physics used for data augmentation into transformer network as opposed to data augmentation. these two sets of methods compete with each other and are worth exploring.
- Training with different context lengths: In table 5, if bold entries are meant to highlight the best performance, then please highlight all entries of equal performance score. Further, correct the wrong hihglights of 0.293 in the GA column, where 0.299 with 7 day window has a higher score.
- Training with alternative predictor models: These are poor baselines and do not support the claim of outcompeting existing methods or reaching state of the art performance. For this, rather, the authors needed to compare against models presented in the studies from related work, and more recent studies in this field.
- Training without exogenous varibales: This is a mostly uniformative ablation. Exciting is that data augmentation works well. However, for the remaining ones, you are removing obviously relevant information from the feature space and demonstrating that SONNET performs better.

Conclusion:
- "Last but not least, the developed techniques, in particular, disaggregation, enhanced Transformer architecture, and physics model-based data augmentation, all have broad applications beyond net load forecasting." True, and they are already extensively used and more advanced, so I wouldn't frame this as a contribution beyond this application, but rather as methods used in different fields that serve this application.

Potential information added to disaggregation that could be added as "exogenous" features to prediction of net load right away, but isn't:
- installed solar capacity C.
- meteorological data
- physical model of solar generation including all specification parameters

**Strengths:**

Originality: The paper introduces a novel approach by applying a disaggregation step in combination with Transformer-based architectures specifically for residual load forecasting, which is less explored in existing literature.

Quality: The use of data augmentation as a means to boost predictive performance adds value and could inspire further innovations in load prediction approaches for renewable energy systems.

Clarity: The structure and presentation are clear, making the methodology and motivation behind each component accessible to readers.

Significance: The work is relevant to the domain of renewable energy forecasting, especially in the context of BTM generation and net load prediction. The potential applicability to other domains is a notable point, even though the methods used are established in related areas.

**Weaknesses:**

Support for Core Claims: The primary claim that disaggregation enhances residual load prediction is insufficiently substantiated. The paper would benefit from additional experiments that compare prediction accuracy with and without the disaggregation step, especially against simpler baseline models that utilize the same information.

Choice of Baselines: The baseline models used in the experiments do not adequately represent the current state-of-the-art (SOTA) methods. A direct comparison with established models from related studies would provide stronger support for claims of outperforming SOTA approaches.

Motivation for High Resolution: The necessity for single-building resolution predictions is unclear, given that aggregated predictions (e.g., at the level of low/medium voltage substations) could suffice for many operational tasks. Clarifying the practical applications of such high-resolution forecasts would strengthen the motivation.

Methodology Choices: There is insufficient discussion on why disaggregation is preferred over simpler approaches, such as directly forecasting net load with relevant exogenous features like solar capacity, meteorological data, and solar generation models. Additionally, the use of data augmentation could be complemented by Physics-Informed Neural Networks (PINNs), which might yield more compact models needing less training data.

Ablation Study Limitations: Although the experiments mention disaggregation, there is limited evidence showing that disaggregation alone significantly improves results. More ablation studies could be provided, including training with and without solar capacity information, to further validate this approach.

**Questions:**

Could the authors clarify why disaggregation is necessary, rather than predicting net load directly with the same exogenous variables used in the disaggregation process?

Can the authors explain why high-resolution forecasts at the level of individual metering points are needed for system operators, as opposed to predictions at a more aggregated level, such as for substations?

What is the intuition behind the expectation that disaggregation would increase performance? If this is related to disentangling feature representations, could the authors discuss this in the context of equivariant feature representation learning?

Have the authors considered using Physics-Informed Neural Networks (PINNs) as an alternative to data augmentation, potentially simplifying the model training requirements?

Could the authors add a comparison with more competitive baseline models and recent methods from the literature to substantiate claims of SOTA performance?

**Details Of Ethics Concerns:**

No direct ethical concerns were identified in this work. However, the approach’s focus on single-building predictions could raise privacy considerations if deployed in residential or commercial areas at such a granular level.

---

### Note · Authors · 2024-11-24

I have read and agree with the venue's withdrawal policy on behalf of myself and my co-authors.